# LED Light Improved by an Optical Filter to Visible Solar-Like Light with High Color Rendering

**Li-Siang Shen [1], Hsing-Yu Wu [2,3,4], Li-Jen Hsiao [2], Chih-Hsuan Shih [1] and Jin-Cherng Hsu [1,5,*]**

[1] Department of Physics, Fu Jen Catholic University, No.510 Zhongzheng Rd., Xinzhuang Dist., New Taipei City 242062, Taiwan; speaker2012012@gmail.com (L.-S.S.); zxc959647833@gmail.com (C.-H.S.)

[2] System Manufacturing Center, National Chung-Shan Institute of Science and Technology, New Taipei City 237209, Taiwan; andy810301@gmail.com (H.-Y.W.); hslijen@gmail.com (L.-J.H.)

[3] Department of Electro-Optical Engineering, National Taipei University of Technology, Taipei 10608, Taiwan

[4] Center for Astronomical Physics and Engineering, National Central University, Taipei 320317, Taiwan

[5] Graduate Institute of Applied Science and Engineering, Fu Jen Catholic University, No. 510 Zhongzheng Rd., Xinzhuang Dist., New Taipei City 242062, Taiwan

\* Correspondence: 054326@mail.fju.edu.tw; Tel.: +886-2-2905-3765

**Abstract:** In this study, a new, cost-effective, rapid, and easy method to produce a sunlight-like D65 light source from a typical white light-emitting diode (LED) is discussed. The novelty of this method is that the emission spectrum of a typical white LED is measured first, then the reverse spectrum is used to design and fabricate a double-sided multilayer coating filter to set in front of the typical white LED. This can be verified experimentally to improve the color-rendering index of the white LED to 95.8 at the D65 color temperature. The optical thicknesses of the multilayer film are designed at a quarter wavelength. The layer-thickness errors during the deposition process are reduced due to easy monitoring with the turning-point method. By lowering both the cost and level of technology required to produce D65 light sources, in addition to the most direct consequences of increased D65 availability and affordability, the cost and level of technology required for research that heavily utilizes D65 light sources can also be lowered in turn, especially in the fields of clinical science, medicine, and related industries.

**Keywords:** LED; sunlight-like light; turning-point method; color-rendering index; D65

## 1. Introduction

Since ancient times, human beings have always been more comfortable and adaptable during the day. Light is essential for the visual perception of our surrounding environment, and affects both our physical and mental conditions through our eyes and skin. Daylight provides an excellent visual environment that benefits our physical and mental health through our eyes and skin, by providing us with clear visual acuity and warmth, as well as in several other ways that do not necessarily involve image-forming effects. However, in certain times and places, for example, in an indoor setting during nighttime, the presence of daylight is absent, and as such humankind has long since developed lighting sources with ever-increasing demands for comfort and safety, from burning wood, wax candles, light bulbs, and fluorescent lamps to the current LED.

Recently, a portable, easy-to-use, and cost-effective D65 light source has been utilized for clinical applications, for example, to illuminate either the pseudoisochromatic plates of the Ishihara color-blind test [1], 16 color caps for the Farnsworth D-15 test [2], and 85 color caps for the Farnsworth-Munsell 100-Hue test [3]. The D65 light is one of the standard illuminants defined by the International Commission on Illumination (CIE). It has a correlated color temperature (CCT) of approximately 6500 K, which corresponds roughly to the average midday light [4]. Figure 1 shows the spectrum of the D65 light. All the tests are evaluated using CIE standardized lighting [1]. This strict examination process

can also apply to medical lighting because LED light sources require a high color-rendering index (CRI) [5]. Its novel miniaturization, portability, robust production process, and small system volume means it has more advantages than conventional xenon arc light sources [6] and fluorescent lamps [7]. The results of the improved D65 LED light are reported in this research.

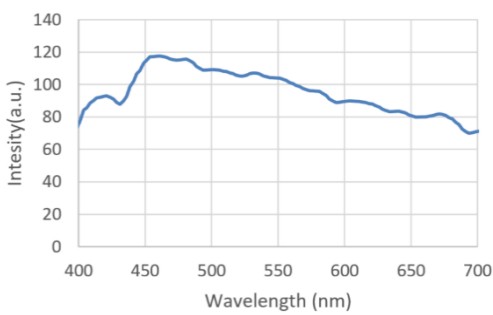

**Figure 1.** The spectrum of D65 light.

Color rendering is one of the essential characteristics of general lighting. It illustrates an object's natural chromaticity under a specific illumination source for developments in general lighting. The color-rendering index of light is the internationally recognized standard to evaluate the color-rendering evaluation index [8]. In this study, the proposed LED light source is compared with a reference light source, D65. The reference light source and the test light source illuminate the eight selected Munsell samples. The color differences ($\Delta E_i$) of each sample illuminated by the two light sources in the CIE 1964 U*V*W* color space can then be calculated [9]. The special color-rendering index ($R_i$) of each color sample is given by the equation

$$R_i = 100 - 4.6\,\Delta E_i\ (i = 1, \ldots , 8), \tag{1}$$

and the score, the average of the eight color samples, is the general color-rendering index.

$$R_a = \sum_{i=1}^{8} \frac{R_i}{8} \tag{2}$$

The perfect color-rendering light score is 100. The $R_a$ value of most LEDs available on the market is only about 80, and this is insufficient to meet the requirements for utilization in clinical science, medicine, and related advanced industries.

D65 is used extensively in many visually demanding applications as a close substitute for natural daylight. With a corrected color temperature (CCT) of 6500 K, D65 is almost equivalent to sunlight during the day at noontime. In recent years, LEDs have become the preferred light source due to their high efficiency, color tunability, durability, long lifetime, energy efficiency, and environmental friendliness [10–12]. A typical white-light LED uses a blue LED (mostly GaN or InGaN) with a central emission peak around 440 to 460 nm to excite a green/yellow phosphorescence and a red phosphorescence inside the sealing transparent silicone resin. The green phosphor has a central emission peak around 500 to 540 nm and the yellow phosphor around 545 to 595 nm. The emission from the blue LED and the excitations of the green/yellow and red phosphorescences then combine to form a pseudo-white-light emission, which is the reason for its name: white-light LED.

This artificially synthesized white light often has a low color-rendering index ($R_a < 80$) and a significantly higher CCT than 4000 K. To address this issue, additional intermediate phosphorescence wavelengths were added to LEDs, and the intensity of the red phosphorescence was increased. With the discovery of highly efficient red-emitting $CaAlSiN_3{:}Eu^{2+}$, $Sr_2Si_5N_8{:}Eu^{2+}$, $K_2SiF_6{:}Mn^{4+}$ or $SrLiAl_3N_4{:}Eu^{2+}$ phosphors with multiple emission centers, the $R_a$ can be improved to reach up to and exceed over 90 [13]. Moreover, developing novel phosphors in the white LED filled the cyan gap between the blue and the yellow emission

in the 470 to 500 nm region, leading to higher-quality general lighting [13,14]. In some patents and documentation, the high color-rendering index is realized using a UV LED to stimulate multiple phosphor-powder mixtures [15], forming multiple LED channels, and then adding a neodymium oxide absorption filter to absorb the overly strong 580 nm emission peak [16]. All efforts enable humans to observe naturally colored objects under more commonly employed LED light sources.

Due to their energy-saving traits, LED light sources will likely see an even broader use in the future, in shops, restaurants, and everyday life in general. This research takes the approach of improving the most common commercially available white-light LED, which has the emission spectrum shown in Figure 2. By adding an optical filter that possesses a reverse transmission spectrum of a typical white-light LED, the intensity distribution of the emission radiation in the 450 to 650 nm range becomes flat and close to that of the D65 light.

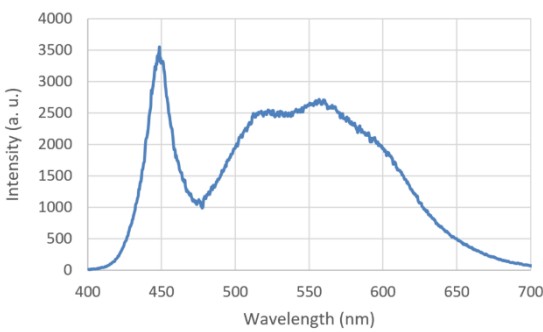

**Figure 2.** The measured emission spectrum of commercial phosphor-based white LED.

## 2. Materials and Methods

### 2.1. Manufacturing of Thin Films

The vacuum coating system is a 90 cm box coater (SGC-120SA-IAD, Showa Shinku CO., Yokohama, Japan) equipped with a 10 kW electron beam gun and an end-Hall ion source (Mark II) made by Veeco Ion Tech. Inc (Plainview, NY, USA). In the deposition process, we use the well-studied coating materials $SiO_2$, $Al_2O_3$, and $Ta_2O_5$ [17–19]. A B270-glass substrate, 36 mm in diameter and 1 mm thick, is loaded onto a substrate holder. The vertical height extending from the circular hearth to the substrate is 75 cm. The horizontal distance from the beam-gun vaporization source to the substrate-holder position is 30 cm. The holder is spun at a rotation rate of 30 rpm to ensure film distribution uniformity during the deposition. The off-axial source coating system is generally used to achieve coating uniformity. The vacuum chamber is pumped down to a base pressure lower than $5 \times 10^{-6}$ Torr, and the substrate is heated to about 220 °C. Before the coating process, the substrate is cleaned using an ion beam for ten minutes with a beam voltage of 120 V and a beam current of about 2.0 A. Oxygen and argon are introduced into the ion source as the working gas by setting the mass flow controllers to 5 and 10 sccm, respectively. An automatic pressure controller is used to control the working pressure to about $8 \times 10^{-5}$ Torr. After setting all of the above deposition parameters, this system can begin depositing the multilayer films onto the substrate. During the multilayer fabrication sequence, the deposition rate is controlled with a quartz monitor at 0.5 nm/s for $SiO_2$ and 0.3 nm/s for $Al_2O_3$ and $Ta_2O_5$. An optical-thickness monitor can almost be operated at the monitoring wavelengths of 450 and 550 nm to control the deposited thickness on both sides of the substrate using the turning-point method [20]. A spectrometer (USB 4000 System, Ocean Optics Inc., Dunedin, FL, USA) is used to measure the spectrum, whose color temperature and $R_a$ value can be determined by self-editing software.

### 2.2. Design of Optical Multilayer Films of Filter

The optical filter of this research is based on Southwell's design of optical notch-filtering multilayer films with coupled wave theory [21]. Ideally, the refractive index of the multilayer stack should be a sine function of the film thickness. However, it is challenging to deposit the film in a sinusoidal refractive index using general electron-gun evaporation due to the index variation in each film. In this research, we fabricated a symmetric-stack notch-filter multilayer in $n$ fundamental periods of $(0.5pq0.5p)^n$, where $p$ and $q$ are odd multiples of quarter-wavelength optical thickness with different refractive indices. The $n$ value controls the transmittance at the central wavelength. The higher the $n$ value, the lower the transmittance. The refractive index of the multilayer stack is a square wave function of the film stack thickness, and is well within the fabrication capability of electron-gun deposition.

The full width of half maximum (FWHM), $2\Delta\lambda$, and the transmittance at the center wavelength ($\lambda = 450$ nm) of the stack are related by an equation [22],

$$2\Delta\lambda = \frac{4\lambda}{\pi} sin^{-1} \frac{n_M - n_L}{n_M + n_L}. \tag{3}$$

As indicated by Equation (3), the smaller the difference between $n_M$ and $n_L$, the smaller the resultant FWHM is. The two selected deposition materials are $Al_2O_3$ and $SiO_2$ at $n_M = 1.671$ and $n_L = 1.465$ at a wavelength of 450 nm, respectively, and the value of $n$ is chosen to be 7 in this research [23]. Moreover, the FWHM of the stack designed by Essential Macleod software (edited by Thin Film Center Inc., Tucson, AZ, USA), with $p = 3M$ and $q = 3L$, is approximately $1/3$ narrower than that of the stack with $p = M$ and $q = L$, as shown in Figure 3. The filter, therefore, was designed by the multilayer structure,

$$Air/(1.5M3L1.5M)^7/Sub/(H'L')^2H'/Air, \tag{4}$$

where M is the quarter-wave optical thickness of $Al_2O_3$, and L is the quarter-wave optical thickness of $SiO_2$ on the top side of the B270 substrate (Sub). H' is the quarter-wave optical thickness of $Ta_2O_5$ ($n_{H'} = 2.141$ at 550 nm wavelength), and L' is the quarter-wave optical thickness of $SiO_2$ ($n_{L'} = 1.459$ at 550 nm wavelength) on the underside.

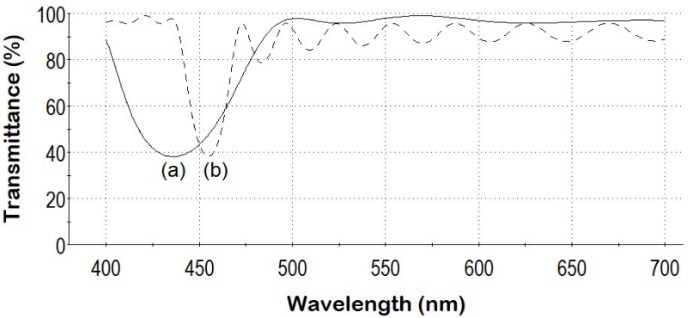

**Figure 3.** (**a**) Notch-filtering multilayer film of $(0.5ML0.5H)^7$ and (**b**) notch-filtering multilayer film of $(1.5M3L1.5H)^7$ are symmetric multilayer stacks designed at a wavelength of 450 nm.

Figure 2 is the measured spectral profile of a typical commercially available white LED, which shows a narrow and intense peak at 450 nm and a broader excited emission centered at 550 nm. To balance the non-uniform intensities, the measured spectra of D65 $\Psi_{D65}$ and the LED $\Psi_{LED}$ are first normalized to their highest intensities. The filter is then designed such that if $\Psi_{LED}$ is passed through the filter with the response $\Psi_{filter}$, the resultant filtered output would be $\Psi_{D65}$. That is,

$$\Psi_{LED} \cdot \Psi_{filter} = \Psi_{D65} \tag{5}$$

can be rearranged to get the following expression

$$\Psi_{\text{filter}} = \Psi_{\text{D65}} / \Psi_{\text{LED}}, \tag{6}$$

which is the target transmittance spectrum of the filter.

### 3. Results

#### 3.1. Filter Design Results and Production

Figure 4 is a plot of the obtained target transmittance spectrum $\Psi_{\text{filter}}$. The desired filter transmittance spectrum is almost an exact reverse of the LED spectrum. Ideally, the LED spectrum would pass through the multilayer filter films. The two characteristic bands would be filtered according to their distinct spectral intensity distribution, leaving the resultant filtered output spectrum close to D65 natural light.

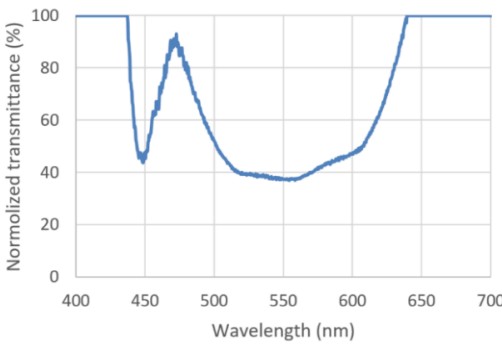

**Figure 4.** The target transmittance spectrum of the filter.

To simplify both the filter design and the multilayer deposition process, two separate notch-filtering multilayer films are deposited on two sides of the substrate to form the combined filter. Designing and fabricating a two-band notch-filtering multilayer film with unequal notch widths on one side of the substrate is significantly more complex than separating the task load to two sides of the substrate, since there is no optical interference between the multilayer films on each side of the "thick" substrate [22]. In addition, the two multilayer designs can independently control the two filtering characteristics.

Figure 3a,b show the design spectra of $(0.5\text{ML}0.5\text{H})^7$ and $(1.5\text{M}3\text{L}1.5\text{M})^7$ notch-filtering multilayer films, respectively. Although their design monitoring wavelengths are all 450 nm, the lowest transmittance wavelength of the $(1.5\text{M}3\text{L}1.5\text{M})^7$ notch-filtering multilayer film is slightly greater than 450 nm, and that of the $(0.5\text{ML}0.5\text{H})^7$ filter is lower. During the coating process of the filter, the monitoring wavelength needs to be adjusted to position the lowest transmittance wavelength at 450 nm. The FWHM of the notch-filtering multilayer film is 33 nm, and the transmittance near 450 nm is 41.4%, as shown in Figure 5.

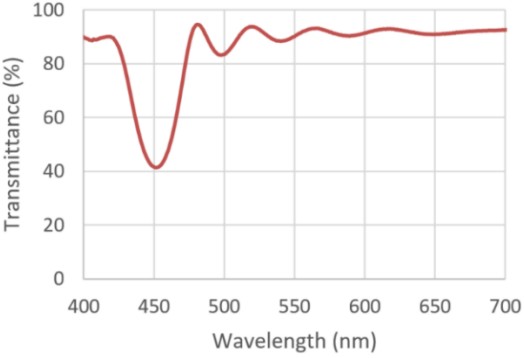

**Figure 5.** The spectrum of the notch-filtering multilayer film on the top side of the substrate.

Table 1 lists the FWHMs and the stack transmittances of the notch-filtering multilayer film (1.5M3L1.5H)[7]. Both the software-simulated values and the actual measured values are listed. The software-simulated FWHM value of 22.4 nm differs significantly from the calculated FWHM value of 12.5 nm by Equation (3). The FWHM calculation using Equation (3) assumes constant refractive indices, which varies from reality since the film's refractive index values of the optical mediums are functions of the wavelength. Generally, the shorter the wavelength is in the visible spectral range, the higher the refractive index. Therefore, the FWHM evaluated by the thin-film software, not disregarding the dispersion property of the films, is larger than the FWHM estimated by Equation (3). The film stack transmittances with seven fundamental periods (i.e., $n = 7$) are 38.2% for the software simulation and 41.4% for the experiment value. The FWHM value by experimental measurement is larger than the simulated value, likely due to the non-uniformness in the distribution of the refractive indices of the deposited films. Figure 5 shows the spectrum of the first fabricated filter, a notch-filtering multilayer film, which reduces the spectral intensity of a specific wavelength band in the range of 440 to 460 nm. The magnitude is dependent on the peak intensity of the 450 nm LED light.

**Table 1.** Evaluations of FWHM and stack (1.5M3L1.5M)[7] transmittance.

| - | Simulated Value [1] | Experiment Value |
|---|---|---|
| FWHM (nm) | 22.4 | 33.0 |
| Transmittance (%) | 38.2 | 41.4 |

[1] Evaluated by optical thin film.

As shown in Figure 2, the FWHM of the yellow phosphorescence excited by the 450 nm light ranges from 490 to 610 nm. The second multilayer deposited on the underside of the substrate has a broader reflection spectrum centered around 550 nm, resulting from the significant refractive index difference between the $n_{H'}$ and $n_{L'}$ of Equation (3). Increasing the number of H'L' layer pairs of Equation (4) can independently decrease the emission of the excess yellow phosphorescence in the spectrum, with the lowest transmittance positioned at 550 nm, as shown in Figure 4. In addition, the transmission spectrum of the $(H'L')^2H'$ multilayer exhibits high transmission at 450 nm. It does not interfere with the 450 nm transmittance of the notch-filtering multilayer film on the top side of the substrate, as shown in Figure 6.

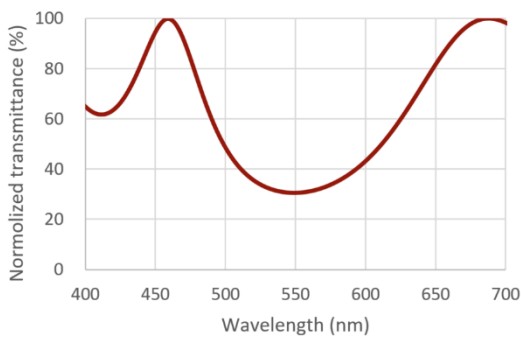

**Figure 6.** The broad spectrum on the second surface of the substrate.

### 3.2. Optical Characteristics of the Modified LED Module

As mentioned in Section 3.1 and shown in Figure 3, the lowest transmittance wavelength of the (1.5M3L1.5M)[7] notch-filtering multilayer film is slightly greater than the design wavelength of 450 nm. The three notch-filtering multilayer films on the topside of the three combined filters are deposited by three monitoring wavelengths somewhat smaller than 450 nm.

Figure 7 shows the transmission spectra of the three combined filters, labeled as $R_a$ 92.4, $R_a$ 93.5, and $R_a$ 95.8, in the visible spectrum of 400 to 700 nm. Due to the broad spectral design, the transmittances around 550 nm of the three multilayer films are almost identical. The lowest transmission wavelength, near 450 nm of the combined filter $R_a$ 95.8, is closer to 450 nm than those of the other two combined filters.

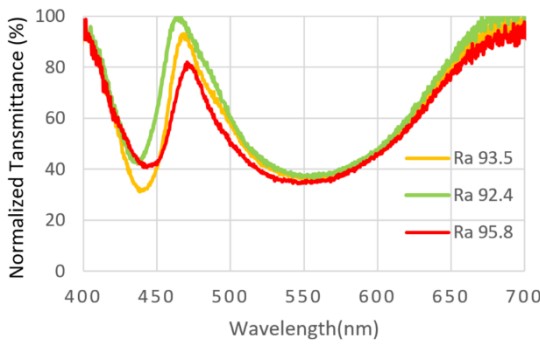

**Figure 7.** The spectra of the three combined filters.

Figure 8 shows the modified LED module integrated with the combined filter in front of the LED cup light. The LED module, by itself, consists of an array of blue LEDs with a narrow emission peak centered around 450 nm as the excitation source, and a green/yellow and red phosphorescence layer to be excited by the emission from the blue LED.

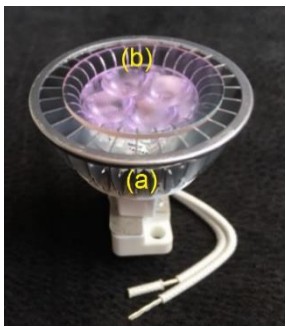

**Figure 8.** Modified LED module integrated with (**a**) 7-watt LED cup light and (**b**) combined filter.

Figure 9 is a plot of the emission spectrum of a typical unfiltered white LED, the emission spectra of the LED after passing through the three fabricated filter samples, and the D65 light spectrum as a reference. The three combined filters reduce the blue 450 nm emission peak and the excess phosphorescence centered around 550 nm, though the cyan gap between the blue and yellow emissions around the 470 to 500 nm region still instigates challenges in enhancing overall color reproduction. However, the combined filters only slightly dampen the intensity around the 475 nm region. At the D65 color temperature, the $R_a$ values of the modified LED module covered with the three combined filters are 92.4, 93.5, and 95.8. The three spectra from 470 to 625 nm almost match that of the D65 light. The fitting degrees of the D65 light spectrum, from 440 to 470 nm, synchronizes with the lowest transmission wavelength near 450 nm of the combined filters. The combined filter $R_a$ 95.8, which has the lowest transmission wavelength near to 450 nm, has the most significant $R_a$ value of 95.8 when covering the modified LED module.

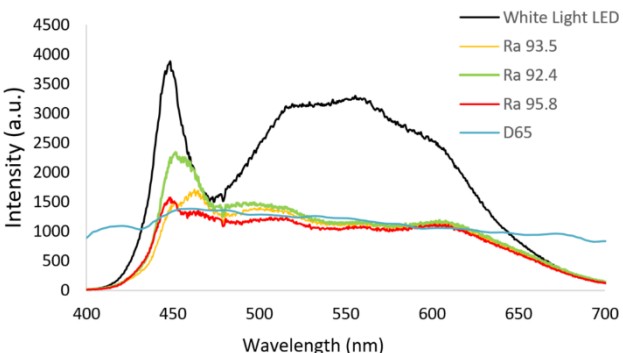

**Figure 9.** The spectra of the white-light LED module, the LED module equipped with the three combined filters, and the D65 light.

Figure 10 is a diagram showing the hue of three wooden boards under three different illumination conditions. Figure 10 shows the illumination from the lights of (a) a 3200 K white LED, (b) the modified LED module with the combined filter $R_a$ 95.8, and (c) a 6000 K white LED, respectively. The boards are painted in a gray 5N level to minimize interference to the hue of the boards under the three illuminations.

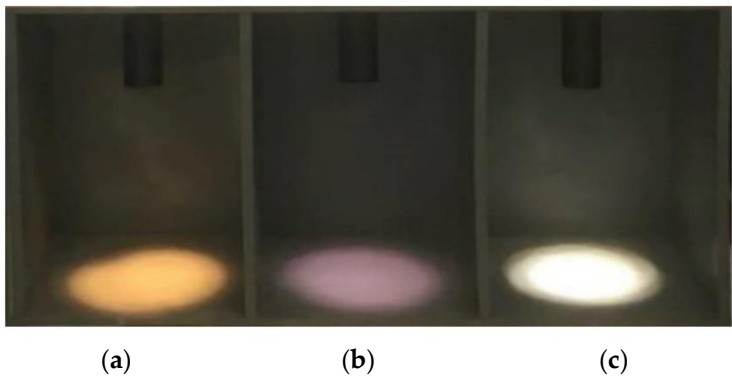

| (a) | (b) | (c) |

**Figure 10.** (**a**) 3200 K LED light source, (**b**) the modified LED module with the combined filter $R_a$ 95.8, and (**c**) 6000 K LED light source to emit three light colors in a three-grid wooden box painted in a gray 5N level. Their $R_a$ values are about 80 at 2800 K color temperature, 95.8 at 6500 K, and 80 at 6000 K, respectively.

The colors appear slightly unnatural under the illumination of the 3200 K and the 6000 K white LED lights. The 3200 K LED illumination slightly accentuates the yellow in the overall tint, and the 6000 K LED illumination exhibits an uncomfortable glare.

## 4. Discussion

With a white LED, the blue 450 nm LED serves both as the excitation source to excite a yellow (and possibly also green and red) phosphorescence and to combine with it to form a "white" light [16]. By utilizing a combined filter with a transmission spectrum in the shape of the inverted spectrum of a typical white LED, the spectral intensity in the overly prominent wavelength bands can be reduced to provide white illumination to objects while retaining their original color. After utilizing the combined filter, the overall output power of the LED was reduced to almost half that of the original unfiltered white LED, though as Žukauskas et al. pointed out, more natural-looking light with better color saturation, which is preferred by most people, often appears duller and mellower [24]. Moreover, according to the study of Babak Zandi et al., human pupil diameter reacts and responds more vitally towards stimuli at longer wavelengths in the visible spectrum; however, with shorter wavelengths towards the blue end of the spectrum, there is little to no change at all. This means that under the same luminosity, the same human-perceived brightness, and

therefore similar eye pupil diameter, light from a typical unfiltered white LED contains a much larger portion of the higher-energy 450 nm blue light, which may cause damage to the retina inside the human eye, since human pupil diameter is less responsive towards blue light. Other than inducing photoreceptor damage, excessive exposure to blue light may also affect a person's physiological functions as well, though it is worth noting that blue light can potentially be used in applications regarding the treatment of certain circadian disorders and sleep-related dysfunctions [25]. Nevertheless, minimizing the risks related to the spectral output of LED-based light sources associated with prolonged exposure to blue light is essential.

An optical filter with the inverted spectrum of a white LED was designed and fabricated in this study. The combined filter was placed in front of a commercially available 6000 K white LED source, which increased the color-rendering index of the white LED from 80 to 95.6 at the D65 color temperature. By being deposited on each side of the BK7 substrate, the combined filter comprised two notch-filter multilayer films. At high incident angles of light, the resultant emission can be observed to undergo a blue shift due to the angular dependency of the filter spectrum. This issue may be mitigated by using curved substrates in the deposition process to decrease the blue shift in future developments. For the scope of this study, however, this issue is solved simply by applying a small aperture to block out the illumination at large angles, as can be observed in Figure 10 by the small circular area of light on the painted boards.

Currently, the most commonly available white LEDs use a combination of blue LEDs and complementary yellow phosphor as the working mechanism. This method has the advantage of structural simplicity and high emission efficiency, and has become the mainstream method of producing white light using LEDs. However, on the other hand, this kind of white-light LED results in higher irradiance towards the shorter wavelengths, more specifically in the blue wavelength region ranging from 450 to 500 nm. This study addressed this issue by designing and fabricating optical filters consisting of multilayer films, which were coated onto both sides of the substrate and acted as limiters. The multilayer film is designed to elicit a specific spectral response through spectrogram analysis of the white-light LED spectrum, such that a smooth resultant illumination spectral intensity profile forms in the visible range between 400 and 700 nm. In the three filter test samples fabricated, the spectrum of the filtered white LED reached as high as 95.8 $R_a$ at the D65 color temperature. Objects under illumination by this filtered white LED can better retain their original natural colors and appearances. The lighting of this kind has high potential in applications involving presenting objects while preserving their actual colors and textures, such as artwork and presentation lightings, retaining the market and artistic value of the artwork with high CRI illumination. Note that for the filter to function properly and increase the $R_a$ value, the color temperature of the selected unfiltered white LED must be higher than 6000 K, because low color temperature LEDs lack the characteristic 450 nm peak. Moreover, to further increase the $R_a$ value, as can be deduced from observing Figure 9, one possible way is to add the lights of 405 and 680 nm LEDs to the mixture light [26].

The utilization of multilayer filters has many advantages: they are easily fabricated, they are well-studied and possess a well-documented design logic, and they have a low fabrication cost. The size of the LED determines the required filter and substrate size, and direct attachment to the hot LED source is also possible due to the generally high heat resistance of the substrate and multilayer materials. The phosphorescence effect may also be simultaneously improved due to the increased back reflection of the 450 nm blue light. This filtering method is safer and less harmful than using UV LEDs and UV-specific phosphors to produce CRI lighting, and is more structurally simplistic than multichannel LEDs. By using the same principles and processes, this filtering method may also increase the already high $R_a$ of an existing LED source to potentially reach 100 $R_a$.

## 5. Conclusions

The double-sided coated filter has two reflective spectral ranges. The aggregated spectrum is intentionally matched to the reverse transmission curve of a commercially available 6000 K white light-emitting diode (LED). When the filter is directly placed in front of the LED light source or module, sunlight-like light will pass through the covered filter. At the D65 color temperature, the general color-rendering index of the filtered light approaches 95.8.

**Author Contributions:** Conceptualization, J.-C.H.; methodology, L.-S.S., J.-C.H. and L.-J.H.; software, L.-S.S. and L.-J.H.; validation, L.-S.S. and L.-J.H.; formal analysis, H.-Y.W.; investigation, H.-Y.W.; resources, J.-C.H. and H.-Y.W.; data curation, C.-H.S.; writing—original draft preparation, J.-C.H. and L.-J.H.; writing—review and editing, J.-C.H. and L.-J.H.; supervision, J.-C.H.; project administration, C.-H.S.; funding acquisition, J.-C.H. and H.-Y.W. All authors have read and agreed to the published version of the manuscript.

**Funding:** This research was funded by the Ministry of Science and Technology of Taiwan, Grant Nos.: MOST 106-2221-E-030-007-MY3 and MOST 106-2112-M-030-001.

**Institutional Review Board Statement:** Not applicable.

**Informed Consent Statement:** Not applicable.

**Data Availability Statement:** The data are included in the article.

**Acknowledgments:** The authors gratefully acknowledge Tzu-Ning Chen for his optical coating technical support.

**Conflicts of Interest:** The authors declare no conflict of interest.

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
