# Peer review of "LED Light Improved by an Optical Filter to Visible Solar-Like Light with High Color Rendering"

_coatings, doi:10.3390/coatings11070763_

Round 1

Reviewer 1 Report

Authors did interesting work, which is improvement of optical properties of commercial LEDs using optical filter. The optical filter was fabricated by depositing Al2O3 and SiO2. Which showed improved Ra f 95.6 at the D65. But authors didn't provide enough evidence of deposition of Al2O3, TiO2 and SiO2.
I have few comments and suggestions to the authors to be conduct before this manuscript to be considered for publication.

  1.  Please provide the material properties and quality of Al2O3, TiO2 and SiO2 source.
  2.  Please provide cross sectional SEM image of deposited Al2O3, TiO2 and SiO2 to confirm the thickness or layered structure. Also if possible TEM image.
  3. Please provide the Auger Electron Spectroscopy (AES) with depth profile to confirm the composition of deposited compounds AlxOy, TiOy and SiOy etc.

Author Response

Response to Reviewer 1 Comments

Authors did interesting work, which is improvement of optical properties of commercial LEDs using optical filter. The optical filter was fabricated by depositing Al2O3 and SiO2. Which showed improved Ra f 95.6 at the D65. But authors didn't provide enough evidence of deposition of Al2O3, TiO2 and SiO2.
I have few comments and suggestions to the authors to be conduct before this manuscript to be considered for publication.

  1.  Please provide the material properties and quality of Al2O3, TiO2 and SiO2 source.
  2.  Please provide cross sectional SEM image of deposited Al2O3, TiO2 and SiO2 to confirm the thickness or layered structure. Also if possible TEM image.
  3. Please provide the Auger Electron Spectroscopy (AES) with depth profile to confirm the composition of deposited compounds AlxOy, TiOy and SiOy etc.

Response: Thanks for your sincere advice. This study uses the well-studied coating materials, Al2O3, Ta2O5, and SiO2 in the deposition process. The four references are cited, as examples:

  • Houska, J.; Blazek, J.; Rezek, J.; Proksova, S. Overview of optical properties of Al2O3 films prepared by various techniques. Thin Solid Films 2012, 520, 5405-5408.
  • Hsu, J.C.; Lee, C.C.; Kuo, C.C.; Chen, S.H.; Wu, J.Y.; Chen, H.L.; Wei, C.Y. Coating uniformity improvement for a dense-wavelength-division-multiplexing filter by use of the etching effect. Appl. Opt. 2005, 44, 4402-4407.
  • Hsu, J.C. Analysis of the thickness uniformity improved by using wire masks for coating optical bandpass filters. Appl. Opt. 2014, 53, 1474-1480.
  • Kumar, M; Kumari, N.; Sharma, A.L.; Karar, V.; Sinha, R.K. Design and fabrication of reflective notch filter using modified thickness modulated Al2O3-SiO2 multilayer. Optical Interference Coatings Conference, New Mexico, USA, 2-7 June 2019, OSA: Washington, D.C., USA, 2019; ThD.6.

The Al2O3, Ta2O5, and SiO2 materials are deposited by ion-assisted electron-beam deposition (IAD). Many deposition parameters, such as temperature, oxygen partial pressure, ion beam current, ion beam voltage during the deposition process, can affect optical qualities. However, the materials are well-studied for many years and usually are used in the general deposition process. Their microstructures, surface roughnesses, and chemical compositions may change, but the optical refractive index and extinction coefficients of the films are stable in the normal deposition process. From the spectrum of the multilayer film, we can know the total optical quality of the three deposited films.

Of course, the material qualities, the cross-sectional SEM image, TEM image, and the Auger Electron Spectroscopy (AES) with depth profile to confirm the composition are also necessary studies for the three films if the research is for UV film, ultra-low film, DWDM film, or high power laser film.

And, We rewrote the abstract  

 “In this study, a new, cost-effective, rapid, and easy method to produce a sunlight-like D65 source from a typical white light-emitting diode (LED) is discussed. The novelty of this method is that the emission spectrum of a typical white LED is first measured; then, the reverse spectrum is used to design and fabricate a double-sided multilayer coating filter to set in front of the typical white LED, which can be verified experimentally to improve the color rendering index of the white LED to 95.8 at the D65 color temperature. The optical thicknesses of the multilayer film are designed at a quarter wavelength. The layer thickness errors during the deposition process are reduced due to easily monitoring through the turning point method. By lowering both the cost and technological requirement to produce D65 light sources, other than the most direct consequences of increased D65 availability and affordability, the cost and technological requirement for researches that heavily utilize D65 sources can also be lowered in turn, especially in the fields of clinical science, medicine, and related industries.”

We have revised the manuscript with red letters.  

Reviewer 2 Report

Shen et al. report the use of an optical filter to achieve high color rendering level with a LED light source. A typical white LED leverages blue light and phosphorescence and is obviously not like a true solar radiation. As demonstrated by the authors, this problem can be addressed by adding an optical filter to tailor the original wavelength spectrum. This work could be interesting for practical applications. However, this work is poor presented. It is also quite difficult for me to define the novelty. The manuscript must be substantially revised before it can be considered for publication in this ("coatings") or other journals.

Specifically, I have the following concerns:

  1. The presentation should be improved. I feel that the current manuscript is more like a student report. The authors may consider to remove many unnecessary information and highlight their contribution in a logic way.
  2. Novelty is low. If my understanding is correct, this article is not at the level of coatings. Overwise, the technology gap should be better defined. In current form, unfortunately, I didn’t see any interesting contributions made by the authors.
  3. Page 7: The figure 8 should be properly labelled, for example, with dashed lines and annotations. Where is the filter? The light purple disk in the center?
  4. Page 9: Author contributions. Initials are incorrect.
  5. Page 9: References are too few. The research background is not properly provided. Introduction should be substantially revised.

Author Response

Response to Reviewer 2 Comments

1. Shen et al. report the use of an optical filter to achieve high color rendering level with a LED light source. A typical white LED leverages blue light and phosphorescence and is obviously not like a true solar radiation. As demonstrated by the authors, this problem can be addressed by adding an optical filter to tailor the original wavelength spectrum. This work could be interesting for practical applications. However, this work is poor presented. It is also quite difficult for me to define the novelty. The manuscript must be substantially revised before it can be considered for publication in this ("coatings") or other journals.  

Response: Thanks for your sincere advice. We revised the manuscript with red letters.

2. Specifically, I have the following concerns:

The presentation should be improved. I feel that the current manuscript is more like a student report. The authors may consider to remove many unnecessary information and highlight their contribution in a logic way.

Response: Thanks for your sincere advice. We have logically rewritten and rearranged many parts of the manuscript.

3. Novelty is low. If my understanding is correct, this article is not at the level of coatings. Overwise, the technology gap should be better defined. In current form, unfortunately, I didn’t see any interesting contributions made by the authors.

Response: Thanks for your sincere advice. In this research, we close the technology gap and can realize Immediately lighting applications in life.

We have revised the manuscript as below:

Abstract:

“In this study, a new, cost-effective, rapid, and easy method to produce a sunlight-like D65 source from a typical white light-emitting diode (LED) is discussed. The novelty of this method is that the emission spectrum of a typical white LED is first measured; then, the reverse spectrum is used to design and fabricate a double-sided multilayer coating filter to set in front of the typical white LED. That is verified experimentally to improve the color rendering index of the white LED to 95.8 at the D65 color temperature. The optical thicknesses of the multilayer film are designed at a quarter wavelength. The layer thickness errors during the deposition process are reduced due to easily monitoring through the turning point method. By lowering both the cost and technological requirement to produce D65 light sources, other than the most direct consequences of increased D65 availability and affordability, the cost and technological requirement for researches that heavily utilize D65 sources can also be lowered in turn, especially in the fields of clinical science, medicine, and related industries.”

Adding some of Introduction

“Recently, a portable, easy-to-use, and cost-effective D65 light source has clinical applications to illuminate either the pseudoisochromatic plates of the Ishihara color-blind test [1], 16 color caps for the Farnsworth D-15 test [2], and 85 color caps for the Farnsworth-Munsell 100-Hue test [3]. All the tests are evaluated by using the CIE standardized lighting [1]. The strictly examining process can also apply to medical lighting because the LED light sources require high color rendering index (CRI) [4]. Its novel miniaturization, portability, robust production process, and small system volume have more advantages than the conventional xenon arc light sources [5] and fluorescent lamps [6]. The results of the improved D65 LED light are reported in this research.”

“The color rendering index of the light is the internationally recognized color rendering evaluation index [7]. A reference light source is Planck radiation with color correlated temperature (CCT) of 6500 K [8] in the study, with the light spectrum shown in Figure 1. The reference light source and the test light source illuminate the eight selected Munsell samples, respectively. The color differences ΔEi, in the CIE 1964 U*V*W* color space, of each sample illuminated by the two light sources are calculated [9]. The special color rendering index (Ri) of each color sample is given by the equation…”

Result:

“As mentioned in section 3.1 and shown in Figure 3, the lowest transmittance wavelength of the (1.5M 3L 1.5M)7 notch filtering multilayer film is slightly greater than the design wavelength of 450 nm. The three notch filtering multilayer films on the topside of the three combined filters are deposited by three monitoring wavelengths somewhat smaller than 450 nm, respectively.”

“Figure 7 shows the transmission spectra of the three the combined filters, labeled by Ra92.4, Ra93.5, and Ra95.8, in the visible spectrum from 400 to 700 nm. Due to the broad spectral design, the transmittances around 550 nm of the three multilayer films are almost identical. The lowest transmission wavelength near 450 nm of the combined filter Ra 95.8 is closer to 450 nm than those of the other two combined filters.”

“At the D65 color temperature, the Ra values of the modified LED module covered with the three combined filters are 92.4, 93.5, and 95.8, respectively. The three spectra from 470 to 625 nm almost match the D65 light. The fitting degrees of the D65 light spectrum from 440 to 470 nm synchronizes with the lowest transmission wavelength near 450 nm of the combined filters. The combined filter Ra95.8, whose lowest wavelength nearest 450 nm, has the most significant Ra value of 95.8 when covering the modified LED module.”

And Conclusion:

“The double-sided coated filter has two reflective spectral ranges. The aggregated spectrum is intentionally matched to the reverse transmission curve of a commercially available 6000 K white light-emitting diode (LED). When the filter is directly placed in front of the LED light source or module, sunlight-like light will pass through the covered filter. At D65 color temperature, the general color rendering index of the filtered light approaches 95.8.”

The added sub-section titles

2.1. Manufacturing of thin films

2.2. Design of optical multilayer films of filter

4. Page 7: The figure 8 should be properly labelled, for example, with dashed lines and annotations. Where is the filter? The light purple disk in the center?

Response: Figure 8 is added two annotations (a) and (b).

5.Page 9: Author contributions. Initials are incorrect.

Response: We did not find the incorrection. Could you please tell us the error of Author contributions that we should correct?  And Thanks. 

6. Page 9: References are too few.

Response: We added the nine references in the manuscript as below:

1.Dain, S.J.; Honson, V.; Curtis, C. Suitability of fluorescent tube light sources for the Ishihara test as determined by colorimetric methods. In Colour Vision Deficiencies XI; Documenta Ophthalmologica Proceedings Series, Sydney, Australia, 21-23 June 1991; Springer: Dordrecht, Netherlands, 1993; p. 327-333.

2. Hovis, J.K.; Neumann, P. Evaluation of light sources for the D-15 color vision test. In Colour Vision Deficiencies XII, Documenta Ophthalmologica Proceedings Series, Tübingen, Germany, 18-22 July 1993; Springer: Dordrecht, Netherlands, 1995; p. 523-529.

3. Racheva, K.; Totev, T.; Natchev, E.; Bocheva, N.; Beirne, R.; Zlatkova, M. Color discrimination assessment in patients with hypothyroidism using the Farnsworth–Munsell 100 hue test. JOSA A 2020, 37, A18-A25.

4. Blaszczak, U.J.; Gryko, L.; Zajac, A. Tunable white light source for medical applications. Proc. SPIE 2017, 10445, p. 104453Y.

5. Powell, L. D65 simulation with a xenon arc. Appl. Opt. 1996, 35, 6708-6713.

6. Teraoka, R.;, Konishi, Y.; Matsuda, Y. Photochemical and oxidative degradation of the solid-state tretinoin tocoferil. Chem. Pharm. Bull. 2001, 49, 368-372.

17. Houska, J.; Blazek, J.; Rezek, J.; Proksova, S. Overview of optical properties of Al2O3 films prepared by various techniques. Thin Solid Films 2012, 520, 5405-5408.

18. Hsu, J.C.; Lee, C.C.; Kuo, C.C.; Chen, S.H.; Wu, J.Y.; Chen, H.L.; Wei, C.Y. Coating uniformity improvement for a dense-wavelength-division-multiplexing filter by use of the etching effect. Appl. Opt. 2005, 44, 4402-4407.

19. Hsu, J.C. Analysis of the thickness uniformity improved by using wire masks for coating optical bandpass filters. Appl. Opt. 2014, 53, 1474-1480.

23. Kumar, M; Kumari, N.; Sharma, A.L.; Karar, V.; Sinha, R.K. Design and fabrication of reflective notch filter using modified thickness modulated Al2O3-SiO2 multilayer. Optical Interference Coatings Conference, New Mexico, USA, 2-7 June 2019, OSA: Washington, D.C., USA, 2019; ThD.6.

7. The research background is not properly provided.

Introduction should be substantially revised.

Response: The research background in the introduction is revised, and the text is revised as follows:

“Recently, a portable, easy-to-use, and cost-effective D65 light source has clinical applications to illuminate either the pseudoisochromatic plates of the Ishihara col-or-blind test [1], 16 color caps for the Farnsworth D-15 test [2], and 85 color caps for the Farnsworth-Munsell 100-Hue test [3]. All the tests are evaluated by using the CIE standardized lighting [1]. The strictly examining process can also apply to medical lighting because the LED light sources require high color rendering index (CRI) [4]. Its novel miniaturization, portability, robust production process, and small system volume have more advantages than the conventional xenon arc light sources [5] and fluorescent lamps [6]. The results of the improved D65 LED light are reported in this research.”

And

“The color rendering index of the light is the internationally recognized color rendering evaluation index [7]. A reference light source is Planck radiation with color correlated temperature (CCT) of 6500 K [8] in the study, with the light spectrum shown in Figure 1. The reference light source and the test light source illuminate the eight selected Munsell samples, respectively. The color differences ΔEi, in the CIE 1964 U*V*W* color space, of each sample illuminated by the two light sources are calculated [9]. The special color rendering index (Ri) of each color sample is given by the equation…”

Reviewer 3 Report

Before evaluating the results presented in the article, the authors need to redo the article in such a way that it is readable. Beginning with the Abstract, the authors wrote that “A typical white-light light-emitting diode (LED) can achieve a sunlight-like spectral profile in the visible spectrum by means of using an optical filter, with an inverted transmission profile to the white LED, fabricated using a deposition coating process.” Apparently, the LED itself can't achieve anything!!! Second, in Abstract, the authors  need to present the main results in a concise form, and not a set of disconnected sentences!

The text of the article should be reworked to exclude such errors from the text as “

calculated in the 1964 W*U*V* uniform color space..”, “ The special color rendering index (CRI, Ri) of each color sample is given by..” and “Air / (1.5M 3L 1.5M)7 / Sub / (H’L' )2 H / Air”,

and so on. Each equation must end with either a comma or a dot.

In the main part of the article, the authors should clearly state what they intend to do, and what has already been done in this area. And yet, it is desirable that the article was edited by a native English speaker, because the form of presentation is simply horrible.

In conclusion, the main results obtained should be presented in a concise form.

The article in the presented form cannot be published in such a highly rated journal as Coatings, and must be reworked in a readable form.

Author Response

Response to Reviewer 3 Comments

  1. Before evaluating the results presented in the article, the authors need to redo the article in such a way that it is readable. Beginning with the Abstract, the authors wrote that “A typical white-light light-emitting diode (LED) can achieve a sunlight-like spectral profile in the visible spectrum by means of using an optical filter, with an inverted transmission profile to the white LED, fabricated using a deposition coating process.” Apparently, the LED itself can't achieve anything!!! Second, in Abstract, the authors need to present the main results in a concise form, and not a set of disconnected sentences!

Response: Thanks for your kind comments. The abstract is re-written as:

“ In this study, a new, cost-effective, rapid, and easy method to produce a sunlight-like D65 source from a typical white light-emitting diode (LED) is discussed. The novelty of this method is that the emission spectrum of a typical white LED is first measured; then, the reverse spectrum is used to design and fabricate a double-sided multilayer coating filter to set in front of the typical white LED. That is verified experimentally to improve the color rendering index of the white LED to 95.8 at the D65 color temperature. The optical thicknesses of the multilayer film are designed at a quarter wavelength. The layer thickness errors during the deposition process are reduced due to easily monitoring through the turning point method. By lowering both the cost and technological requirement to produce D65 light sources, other than the most direct consequences of increased D65 availability and affordability, the cost and technological requirement for researches that heavily utilize D65 sources can also be lowered in turn, especially in the fields of clinical science, medicine, and related industries.”

2. The text of the article should be reworked to exclude such errors from the text as “… calculated in the 1964 W*U*V* uniform color space..”,“ The special color rendering index (CRI, Ri) of each color sample is given by..”

Response:

“The color rendering index of the light is the internationally recognized color rendering evaluation index [7]. A reference light source is Planck radiation with color correlated temperature (CCT) of 6500 K [8] in the study, with the light spectrum shown in Figure 1. The reference light source and the test light source illuminate the eight selected Munsell samples, respectively. The color differences ΔEi, in the CIE 1964 U*V*W* color space, of each sample illuminated by the two light sources are calculated [9]. The special color rendering index (Ri) of each color sample is given by the equation…”

3. and “Air / (1.5M 3L 1.5M)7 / Sub / (H’L' )2 H / Air”,

Response: 

“Air / (1.5M 3L 1.5M)7 / Sub / (H’L' )2 H / Air” is rewritten to “Air / (1.5M 3L 1.5M)7 / Sub / (H’L’ )2 H’ / Air”

All errors in the manuscript have been corrected and displayed in red.

4. and so on. Each equation must end with either a comma or a dot.

Response: Each equation ended with either a comma or a dot in your submitted manuscript. The re-edited manuscript deleted the end comma or a dot. We have re-written each equation with the comma or dot.

5. In the main part of the article, the authors should clearly state what they intend to do, and what has already been done in this area.

Response: We added the following parts:

Introduction:

“Recently, a portable, easy-to-use, and cost-effective D65 light source has clinical applications to illuminate either the pseudoisochromatic plates of the Ishihara color-blind test [1], 16 color caps for the Farnsworth D-15 test [2], and 85 color caps for the Farnsworth-Munsell 100-Hue test [3]. All the tests are evaluated by using the CIE standardized lighting [1]. The strictly examining process can also apply to medical lighting because the LED light sources require high color rendering index (CRI) [4]. Its novel miniaturization, portability, robust production process, and small system volume have more advantages than the conventional xenon arc light sources [5] and fluorescent lamps [6]. The results of the improved D65 LED light are reported in this research.”

6. And yet, it is desirable that the article was edited by a native English speaker, because the form of presentation is simply horrible.

Response: We have revised the manuscript by a native English speaker, as shown in red letters in the manuscript.

7. In conclusion, the main results obtained should be presented in a concise form.

Response: We re-wrote the conclusion as below:

 “The double-sided coated filter has two reflective spectral ranges. The aggregated spectrum is intentionally matched to the reverse transmission curve of a commercially available 6000 K white light-emitting diode (LED). When the filter is directly placed in front of the LED light source or module, sunlight-like light will pass through the covered filter. At D65 color temperature, the general color rendering index of the filtered light approaches 95.8”

8. The article in the presented form cannot be published in such a highly rated journal as Coatings, and must be reworked in a readable form.

Response: Thanks for your sincere advice. We re-wrote the manuscript of abstract, introduction, and conclusion. as follows:

Abstract: “In this study, a new, cost-effective, rapid, and easy method to produce a sunlight-like D65 source from a typical white light-emitting diode (LED) is discussed. The novelty of this method is that the emission spectrum of a typical white LED is first measured; then, the reverse spectrum is used to design and fabricate a double-sided multilayer coating filter to set in front of the typical white LED. That is verified experimentally to improve the color rendering index of the white LED to 95.8 at the D65 color temperature. The optical thicknesses of the multilayer film are designed at a quarter wavelength. The layer thickness errors during the deposition process are reduced due to easily monitoring through the turning point method. By lowering both the cost and technological requirement to produce D65 light sources, other than the most direct consequences of increased D65 availability and affordability, the cost and technological requirement for researches that heavily utilize D65 sources can also be lowered in turn, especially in the fields of clinical science, medicine, and related industries.”

Adding some Introduction

Recently, a portable, easy-to-use, and cost-effective D65 light source has clinical applications to illuminate either the pseudoisochromatic plates of the Ishihara col-or-blind test [1], 16 color caps for the Farnsworth D-15 test [2], and 85 color caps for the Farnsworth-Munsell 100-Hue test [3]. All the tests are evaluated by using the CIE standardized lighting [1]. The strictly examining process can also apply to medical lighting because the LED light sources require high color rendering index (CRI) [4]. Its novel miniaturization, portability, robust production process, and small system volume have more advantages than the conventional xenon arc light sources [5] and fluorescent lamps [6]. The results of the improved D65 LED light are reported in this research.

“The color rendering index of the light is the internationally recognized color rendering evaluation index [7]. A reference light source is Planck radiation with color correlated temperature (CCT) of 6500 K [8] in the study, with the light spectrum shown in Figure 1. The reference light source and the test light source illuminate the eight selected Munsell samples, respectively. The color differences ΔEi, in the CIE 1964 U*V*W* color space, of each sample illuminated by the two light sources are calculated [9]. The special color rendering index (Ri) of each color sample is given by the equation…”

Results:

“As mentioned in section 3.1 and shown in Figure 3, the lowest transmittance wavelength of the (1.5M 3L 1.5M)7 notch filtering multilayer film is slightly greater than the design wavelength of 450 nm. The three notch filtering multilayer films on the topside of the three combined filters are deposited by three monitoring wavelengths somewhat smaller than 450 nm, respectively.”

“Figure 7 shows the transmission spectra of the three the combined filters, labeled by Ra92.4, Ra93.5, and Ra95.8, in the visible spectrum from 400 to 700 nm. Due to the broad spectral design, the transmittances around 550 nm of the three multilayer films are almost identical. The lowest transmission wavelength near 450 nm of the combined filter Ra 95.8 is closer to 450 nm than those of the other two combined filters.”

“At the D65 color temperature, the Ra values of the modified LED module covered with the three combined filters are 92.4, 93.5, and 95.8, respectively. The three spectra from 470 to 625 nm almost match the D65 light. The fitting degrees of the D65 light spectrum from 440 to 470 nm synchronizes with the lowest transmission wavelength near 450 nm of the combined filters. The combined filter Ra95.8, whose lowest wavelength nearest 450 nm, has the most significant Ra value of 95.8 when covering the modified LED module.”

And Conclusion

“A double-sided coated filter has two reflective spectral ranges. The aggregate spectrum intentionally matches the reverse transmission curve of a commercially available 6000 K white light-emitting diode (LED). When the filter is directly placed in front of the LED source or module, sunlight-like light is through the covered filter. The general color rendering index of the filtered light approaches 95.6 at the D65 color temperature.”

We remove the below text in Results to Introduction

“Figure 2 is the measured spectral profile of a typical commercially available white LED, which shows a narrow and intense peak at 450 nm, and a broader excited emission centered at 550 nm. To be balanced the non-uniform intensities, the measured spectra of D65 ΨD65 and the LED ΨLED are first normalized to their highest intensities, respectively. The filter is then designed such that if ΨLED is passed through the filter with the response Ψfilter, the resultant filtered output would be ΨD65. That is,

ΨLED Ψfilter = ΨD65,                                                                             (5) can be rearranged to get the following expression

Ψfilter = ΨD65 / ΨLED,                                                                           (6) which is the target transmittance spectrum of the filter.”

We remove the below text in Results to Discussion

“With a white LED, the blue 450 nm…………. exposure to blue light is essential.”

But we necessarily added the below text :

“At the D65 color temperature, the Ra values of the modified LED module covered with the three combined filters are 92.4, 93.5, and 95.8, respectively. The three spectra from 470 to 625 nm almost match the D65 light. The fitting degrees of the D65 light spectrum from 440 to 470 nm synchronizes with the lowest transmission wavelength near 450 nm of the combined filters. The combined filter Ra95.8, whose lowest wavelength nearest 450 nm, has the most significant Ra value of 95.8 when covering the modified LED module.” 

“The color appears slightly unnatural under the illumination of the 3200-K and the 6000-K white LED lights. The 3200-K LED illumination slightly accentuates the yellow in the overall tint, and the 6000-K LED illumination exhibits an uncomfortable glare, respectively.”

Round 2

Reviewer 1 Report

Thanks to the authors for their reply, though it was better to include the film properties. Any way including the references describing film deposition and characterization is ok.

Author Response

Thanks for your comment.

Reviewer 2 Report

The authors have made substantial revisions to the manuscript. I would recommend the publication as it is.

The only remaining concern is on the author initials. For example, who is "J.C."? Do you mean " Jin-Cherng Hsu"? Then, it should be J.H. or J.-C. H. The same for other author initials.

Author Response

Thanks for your comment.

The only remaining concern is on the author initials. For example, who is "J.C."? Do you mean " Jin-Cherng Hsu"? Then, it should be J.H. or J.-C. H. The same for other author initials.

Response: Thanks for the explanation, we have revised all Author Contributions.

Reviewer 3 Report

The revised version of the article fully meets all the requirements for manuscripts submitted for publication in Coatings. In this study a new and cost-effective method to produce a sunlight-like D65 source from a typical white light-emitting diode (LED) is discussed. A feature of this method is that the emission spectrum of a typical white LED is first measured, and then the reverse spectrum is used to design and manufacture a double-sided multi-layer coating filter to be installed in front of a typical white LED. The optical thickness of the multilayer film is designed for a quarter of the wavelength, and the layer thickness errors during the deposition process are reduced due to the ease of control using the pivot point method.

The paper shows that by reducing both the costs and the technological requirements for the production of D65 light sources, the costs and technological requirements for research in which D65 sources are widely used can be reduced, especially in the field of clinical science, medicine and related industries.

At the beginning of the article, it is necessary to clarify what it means the D65 source!

After this clarification, the article can be published in Coatings.

Author Response

Thanks for your comment.

At the beginning of the article, it is necessary to clarify what it means the D65 source!

Response:

“The D65 light is one of the standard illuminants defined by the international commission on illumination (CIE). It has a correlated color temperature (CCT) of approximately 6500 K and corresponds roughly to the average midday light [4]. Figure 1 shows the spectrum of the D65 light.” was added to line 44.

Line 56: “The proposed LED light source is compared to a reference light source emitted from the Planck radiation with a color correlated temperature of 6500 K [8], as shown in Figure 1.” is revised to “In this study, the proposed LED light source is compared to a reference light source D65.”

We have changed the reference [8] to [4], and [4] to [5], [5] to [6]…..[7] to[8].

We have changed the reference [8] to [4], and [4] to [5], [5] to [6]…..[7] to[8].